# STEM: Scaling Transformers with Embedding Modules

**Ranajoy Sadhukhan**[†]**, Sheng Cao**[§]**, Harry Dong**[†]**, Changsheng Zhao**[§]
**Attiano Purpura-Pontoniere**[§]**, Yuandong Tian**[§]**, Zechun Liu**[§]**, Beidi Chen**[†]
[†]Carnegie Mellon University        [§]Meta AI

## Abstract

Fine-grained sparsity promises higher parametric capacity without proportional per-token compute, but often suffers from training instability, load balancing, and communication overhead. We introduce **STEM** (*Scaling Transformers with Embedding Modules*), a static, token-indexed approach that replaces the FFN up-projection with a layer-local embedding lookup while keeping the gate and down-projection dense. This removes runtime routing, enables CPU offload with asynchronous prefetch, and decouples capacity from both per-token FLOPs and cross-device communication. Empirically, STEM trains stably despite extreme sparsity. It improves downstream performance over dense baselines while reducing per-token FLOPs and parameter accesses (eliminating roughly one-third of FFN parameters). STEM learns embedding spaces with large angular spread which enhances its knowledge storage capacity. In addition, STEM strengthens long-context performance: as sequence length grows, more distinct parameters are activated, yielding practical test-time capacity scaling. Across 350M and 1B model scales, STEM delivers up to $\sim$3–4% improvements in average downstream performance, with notable gains on knowledge and reasoning-heavy benchmarks (ARC-Challenge, OpenBookQA, GSM8K, MMLU). Overall, STEM is an effective way of scaling parametric memory while remaining simpler to train and deploy than existing fine-grained sparse models.

## 1 Introduction

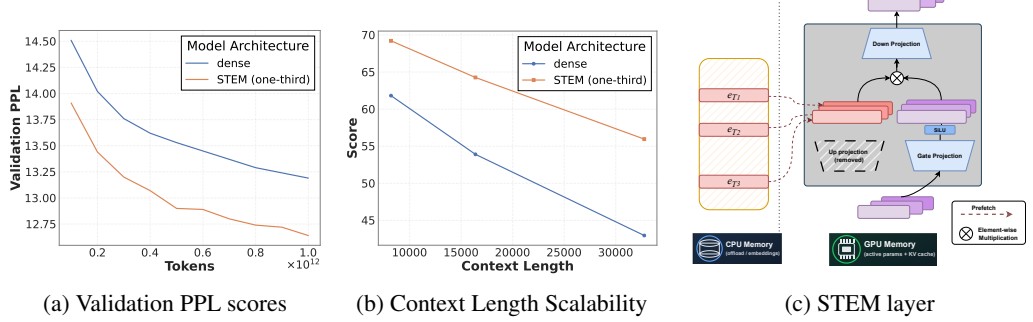

| (a) Validation PPL scores | (b) Context Length Scalability | (c) STEM layer |
| --- | --- | --- |

Figure 1: (a) Validation PPL vs. training tokens for 1B STEM vs. dense; (b) Needle-in-a-Haystack at 8k/16k/32k; (c) STEM layer: embedding tables offloaded to CPU and token-indexed ones are prefetched to GPU.

Sparse computation is a key mechanism for realizing the benefits predicted by parameter-scaling laws (Kaplan et al., 2020; Hoffmann et al., 2022) without proportionally increasing per-token compute. In particular, Mixture-of-Experts (MoE) (Shazeer et al., 2017; Artetxe et al., 2022; Fedus et al., 2022) models have been adopted in several frontier LLMs (Team, 2025b;a; Dai et al., 2024) because they raise *parametric capacity* at roughly constant *activated* FLOPs by sparsely activating a small subset of experts per token. Recent work (Boix-Adsera & Rigollet, 2025; He, 2024; Databricks,

2024; Dai et al., 2024) further advocate for *finer-grained* sparsity that employs large number of *micro-experts* to achieve better expressivity, enhanced knowledge storing capacity, and favorable efficiency metrics.

However, finer granularity introduces nontrivial challenges in both optimization and systems. On the training side, even large fraction of experts can remain under-trained (Huang et al., 2025) due to a highly non-uniform routing and result in training instability. While load-balancing objectives (Shazeer et al., 2017; Fedus et al., 2022; Lepikhin et al., 2020) can address these issues, they may interfere with the primary objective if not carefully tuned (Dai et al., 2024; Qiu et al., 2025; Go & Mahajan, 2025). On the systems side, increasing the number of experts typically raises the number of all-to-all messages while shrinking message sizes, degrading bandwidth utilization and amplifying communication overhead (Huang et al., 2024; Li et al., 2025b). Finer granularity can also reduce parameter-access locality and degrade kernel efficiency when expert subnetworks become too small for dense linear-algebra kernels to reach high occupancy, yielding suboptimal end-to-end performance. To harness the full potential of fine-grained sparsity, we require: **(a)** *stable optimization*, **(b)** *broad expert utilization* (each micro-expert learns useful representations), and **(c)** *negligible expert-retrieval latency and communication overhead*.

We identify static sparsity as a potential solution to achieve these desired properties. Static sparsity keeps the compute path predictable (no runtime routing latency), enables prefetch and CPU offloading (removing the need for inter-node communication). Recently, static sparsity via token-indexed routing has emerged as a promising direction (Roller et al., 2021; Google DeepMind, 2024) with strong performance guarantees. However, such token-based selection strategy lacks context adaptivity. If applied naively, it can reduce the expressivity of the model and degrade quality despite more parameters. Our ablation study in sec. 4.4.3 highlights the criticality of selecting the suitable module for sparsification.

Based on these observations, we introduce *STEM*, a static, token-indexed, fine-grained mechanism that replaces *only* the up-projection in gated FFNs with a token-specific vector retrieved from a layer-local embedding table. The gating and down-projection paths are preserved and shared across tokens. We observe that STEM achieves the following:

*Better Training Stability:* Despite being extremely sparse, STEM does not exhibit any training instability issues as usually seen in MoE models. Figure 3a shows that unlike MoE models, STEM does not exhibit any loss spikes.

*Improved Performance with Larger Knowledge Capacity:* STEM learns a representation space for the embeddings that is conducive to better information storage. The learned embeddings exhibit a large angular spread (i.e., low pairwise cosine similarity), which reduces representational interference and improves addressability of the parametric memory. As a result, it effectively increases the distinct "slots" available for storing and retrieving information. In our downstream evaluation benchmark, STEM consistently outperforms the dense baseline on knowledge-intensive tasks like, ARC-Challenge (Clark et al., 2018), and OpenBookQA (Mihaylov et al., 2018) by large margins (~9–10%).

*Improved Long-context Inference:* During long-context inference, STEM activates more distinct parameters as sequence length grows, yielding test-time capacity scaling. As shown in Figure 1b, the benefits strengthen with context: on Needle-in-a-Haystack (NIAH) (Kamradt, 2024), the gap over the dense baseline increases from 8.4% to 13%.

*Training and Inference-time efficiency:* STEM reduces both FLOPs as well as parameter loading cost by eliminating one-third of the parameters in FFN layers. Consequently, it is strictly more efficient during both computation-intensive training and prefilling, as well as in memory-intensive decoding.

We benchmark STEM against the dense baseline with 350M MobileLLM (Liu et al., 2024) and Llama3.2-1B (Meta AI, 2024) model variants. Additionally, we compare with Hash Layer MoEs with the *same total parameter count*. We report results on standard downstream suites across pre-training, mid-training, and context-length extension. STEM improves downstream accuracy by up to ~3–4% while reducing per-token FLOPs and parameter accesses by up to one-third. It also strengthens knowledge retrieval and mathematical reasoning, with gains on GSM8K (Cobbe et al.,

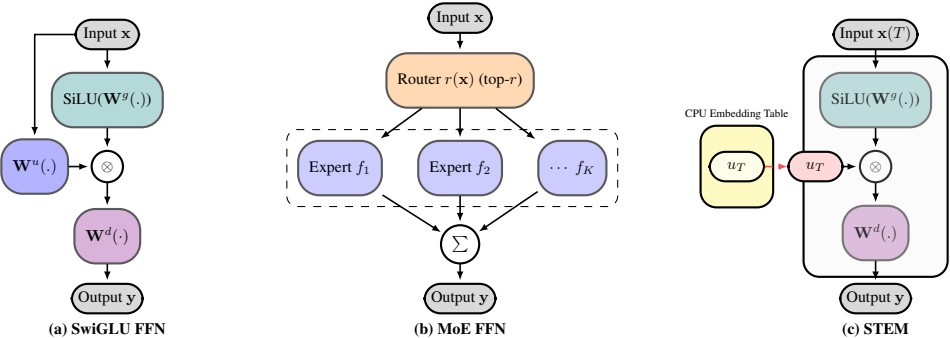

Figure 2: Schematics of (a) SwiGLU FFN, (b) MoE FFN, and (c) STEM with a single prefetched token embedding. In MoE FFN, the full FFN module is considered as one expert.

2021) and MMLU (Hendrycks et al., 2021), and shows pronounced improvements on Needle-in-a-Haystack (Kamradt, 2024) at longer contexts.

## 2 METHOD

### 2.1 BACKGROUND

Consider a decoder-only transformer with $N$ layers, vocabulary size $V$, model width $d$, and feed-forward width $d_{\text{ff}}$. For a given layer $\ell$, the SwiGLU feed-forward block uses a gate projection $\mathbf{W}_\ell^g \in \mathbb{R}^{d_{\text{ff}} \times d}$, an up projection $\mathbf{W}_\ell^u \in \mathbb{R}^{d_{\text{ff}} \times d}$, and a down projection $\mathbf{W}_\ell^d \in \mathbb{R}^{d \times d_{\text{ff}}}$. Consider, $t \in \{1, \ldots, V\}$ denote the vocabulary id of the current token, and the corresponding input hidden state of the $\ell^{th}$ FFN layer is given by $\mathbf{x}_\ell \in \mathbb{R}^d$. Then the transformation in the FFN layer is

$$\mathbf{y}_\ell = \mathbf{W}_\ell^d \big( \text{SiLU}(\mathbf{W}_\ell^g \mathbf{x}_\ell) \odot (\mathbf{W}_\ell^u \mathbf{x}_\ell) \big), \tag{1}$$

where $\odot$ denotes elementwise multiplication.

**Mixture-of-Experts (MoE).** In MoE, a dense FFN is replaced by $K$ expert FFNs $\{f_{\ell,k}\}_{k=1}^K$ and a router $r_\ell(\mathbf{x}_\ell)$ that selects $\mathcal{T}_\ell(\mathbf{x}_\ell)$ (top-$r$ experts) with mixture weights $\pi_{\ell,k}(\mathbf{x}_\ell)$ (Artetxe et al., 2022; Fedus et al., 2022). With SwiGLU experts,

$$f_{\ell,k}(\mathbf{x}_\ell) := \mathbf{W}_{\ell,k}^d \big( \text{SiLU}(\mathbf{W}_{\ell,k}^g \mathbf{x}_\ell) \odot (\mathbf{W}_{\ell,k}^u \mathbf{x}_\ell) \big), \quad \mathbf{W}_{\ell,k}^d \in \mathbb{R}^{d \times d_{\text{ff}}},$$

the layer output is

$$\mathbf{y}_\ell = \sum_{k \in \mathcal{T}_\ell(\mathbf{x}_\ell)} \pi_{\ell,k}(\mathbf{x}_\ell) \, f_{\ell,k}(\mathbf{x}_\ell) \tag{2}$$

**Token-indexed Mixture-of-Experts.** To eliminate the routing parameters and auxiliary routing loss functions, (Roller et al., 2021) fixed mapping from input token ids to experts based on random and balanced hash functions. Consequently, the FFN output is computed as,

$$\mathbf{y}_\ell = \sum_{k \in \text{hash}(t)} f_{\ell,k}(\mathbf{x}_\ell) \tag{3}$$

### 2.2 STEM

Unlike MoE alternatives, STEM only replaces the dense up-projection in the SwiGLU FFN with a *token-indexed* vector looked up from a per-layer table. For layer $\ell$, let $\mathbf{U}_\ell \in \mathbb{R}^{V \times d_{\text{ff}}}$ be the embedding table. Given input $\mathbf{x}_\ell \in \mathbb{R}^d$, the STEM layer computes

$$\mathbf{y}_\ell = \mathbf{W}_\ell^d \big( \text{SiLU}(\mathbf{W}_\ell^g \mathbf{x}_\ell) \odot \mathbf{U}_\ell[t] \big), \tag{4}$$

where $\mathbf{U}_\ell[t] \in \mathbb{R}^{d_{\text{ff}}}$ is the row of $\mathbf{U}_\ell$ corresponding to token $t$ and $\odot$ denotes elementwise multiplication. We provide a simple schematic diagram for dense baseline (SwiGLU FFN), MoE and STEM in Fig 2.

Table 1: Theoretical efficiency for each decoder FFN layer when replacing the FFN up-projection with a token-indexed STEM embedding table. We assume SwiGLU, ignore biases, and count elementwise ops as $\mathcal{O}(DL)$.

| | **FFN** | **STEM** | Savings ($\Delta$) |
|---|---|---|---|
| *Prefill / training (batch size B, sequence length L)* | | | |
| FLOPs | $B(3d_{\text{ff}}dL + d_{\text{ff}}L)$ | $B(2d_{\text{ff}}dL + d_{\text{ff}}L)$ | $B(dd_{\text{ff}}L)$ |
| Communication | $0$ | $\text{uniq}(BL)d_{ff}$ | |
| *Decoding (per step, batch size B)* | | | |
| Parameter loading cost | $3dd_{\text{ff}}$ | $2dd_{\text{ff}}$ | $dd_{\text{ff}}$ |
| Communication | $0$ | $B_{\text{uniq}}d_{ff}$ | |

**Notation:** $d$: model width; $D$: FFN hidden size; $L$: context length; $L_{\text{uniq}}$: number of unique tokens in the $L$-token context; $B_{\text{uniq}}$: number of unique tokens across the batch at a decode step ($\leq B$); $\text{uniq}(BL)$: number of unique tokens across the $BL$ tokens in a training batch.
**Notes:** Training multiplies both FLOP counts by $\approx$ the usual forward+backward factor, but the saving $\Delta\text{FLOPs} = dDL$ remains. Communication doubles during training as gradients of the STEM embeddings are transferred back to CPU for optimizer update.

## 2.3 STEM[†]

STEM uses strictly fewer active parameters, and FLOPs for each token. And because of the architectural bias, STEM is susceptible to some loss of contextual learning ability. We also introduce a hybrid variant of STEM, which retains the up projection matrix in FFN, but complements with an additive token-specific modulation. Concretely, the new variant STEM[†] computes the FFN output as follows,

$$\mathbf{y}_\ell = \mathbf{W}_\ell^d\big(\text{SiLU}(\mathbf{W}_\ell^g\mathbf{x}_\ell) \odot (\mathbf{W}_\ell^u\mathbf{x}_\ell + \mathbf{U}_\ell[t])\big), \tag{5}$$

# 3 ANALYSIS

## 3.1 EFFICIENCY

STEM improves both computation and memory access. During compute-intensive phases (training and prefill), replacing the FFN up-projection with token-indexed embeddings reduces the per-layer FLOPs. During memory-intensive decoding, it lowers parameter traffic relative to a dense up-projection. Table 1 summarizes the per-layer counts and the resulting savings. Below we present a simple theoretical analysis of the training and inference efficiency for a *single* decoder layer.

**Training efficiency.** Consider a batch of $B$ sequences with sequence length $L$, hidden width $d$, and FFN hidden size $d_{\text{ff}}$. Ignoring elementwise ops and biases, the per-layer training FLOPs (forward + backward + weight gradients) can be written as

$$F_{\text{train}}^{\text{base}} = B\big(4Ld^2 + 2L^2d + 3Ld\,d_{\text{ff}}\big),$$
$$F_{\text{train}}^{\text{stem}} = B\big(\underbrace{4Ld^2 + 2L^2d}_{\text{Attn}} + \underbrace{2Ld\,d_{\text{ff}}}_{\text{MLP}}\big).$$

The per-layer FLOPs reduction of STEM is therefore

$$\Delta F_{\text{train}} = F_{\text{train}}^{\text{base}} - F_{\text{train}}^{\text{stem}} = BLd\,d_{\text{ff}},$$

and the corresponding saving fraction is

$$\text{saving fraction} = \frac{\Delta F_{\text{train}}}{F_{\text{train}}^{\text{base}}} = \frac{d_{\text{ff}}}{4d + 2L + 3d_{\text{ff}}}.$$

Plugging in the architecture hyperparameters for each Qwen2.5 model yields saving fractions of $21.7\%$ for Qwen2.5-1.5B, $22.8\%$ for Qwen2.5-3B, $23.9\%$ for Qwen2.5-7B, $19.7\%$ for Qwen2.5-14B, and $24.8\%$ for Qwen2.5-32B.

**Inference efficiency.** Prefill efficiency closely matches training efficiency because both are compute-bound. In contrast, decoding is primarily memory-bound: the dominant cost is loading parameters and KV cache rather than doing FLOPs. For a batch size $B$ and context length $L$, we can write the per-layer memory access cost as

$$M_{\text{dec}}^{\text{base}} = B\left(4d^2 + 2Ld + 3d\,d_{\text{ff}}\right),$$

$$M_{\text{dec}}^{\text{stem}} = B\big(\underbrace{2Ld}_{\text{KV cache}} + \underbrace{4d^2 + 2d\,d_{\text{ff}}}_{\text{projection params}}\big).$$

The reduction in parameter loading cost is

$$\Delta M_{\text{dec}} = M_{\text{dec}}^{\text{base}} - M_{\text{dec}}^{\text{stem}} = Bd\,d_{\text{ff}},$$

so the saving fraction is

$$\text{saving fraction} = \frac{\Delta M_{\text{dec}}}{M_{\text{dec}}^{\text{base}}} = \frac{d_{\text{ff}}}{4d + 2L + 3d_{\text{ff}}},$$

which matches the FLOPs saving factor during training and prefill. As the batch size grows, the linear layers become increasingly compute-bound, and STEM's per-layer FLOPs reduction ensures that this efficiency gain is sustained even in the high-throughput regime.

A key difference from MoE is how cost scales with batch size. In STEM, parameter traffic grows mainly with the number of unique tokens seen. In contrast, MoE expert selection expands with batch size and routing diversity; larger batches tend to light up more experts, quickly eroding the sparsity benefit.

### 3.1.1 VRAM AND COMMUNICATION SAVINGS

MoE models use a lot of VRAM. The expert subnetworks must stay on the GPU, or be fetched repeatedly. Expert parallelism also needs all-to-all communication, even when only a few experts are active (Huang et al., 2024; Go & Mahajan, 2025). STEM avoids these costs. Its embeddings are token-indexed and local to each layer, so the model can prefetch them without any routing logic. These tables are separate from the matmul weights, so we can offload them to CPU memory. In our setups, this frees up roughly one-third of the FFN parameter memory. We can also replicate the embedding tables in CPU memory on every serving node. This eliminates cross-node expert traffic and the synchronization overhead of expert parallelism.

**Prefetching cost.** The prefetching cost can be greatly reduced by deduplicating the STEM embeddings of the batched tokens. We can further cut traffic by caching the most frequently used STEM embeddings, using the extra memory we save from removing the up-projection matrices. As the model embedding size grows, compute cost increases quadratically, but prefetching cost grows only linearly. This makes CPU-offloaded STEM increasingly attractive and scalable for larger model sizes.

### 3.2 CONTEXT-LENGTH ADAPTIVE PARAMETER USAGE

Because STEM employs token-indexed, fine-grained sparsity, the number of *distinct* parameters touched in a forward pass grows with the number of *unique* tokens in the window. Aside from the shared projections in attention (Q/K/V/O) and the gated FFN's gate/down projections, the STEM module draws one vector per token ID per layer; repeated tokens reuse the same vector, while novel tokens activate new ones. Let $L$ be the context length and $L_{uniq}$ the count of unique token ids in the sequence; with STEM applied at layers $\mathcal{S}$ and FFN width $d_{\text{ff}}$, the STEM-specific parameters *activated* by a single sequence are

$$\text{Params}_{\text{act}}^{\text{STEM}}(L) = |\mathcal{S}|\,d_{\text{ff}}\,L_{uniq}.$$

In natural text $L_{uniq}$ typically grows sublinearly (Heaps-like), so longer contexts steadily engage more parameters without increasing per-token FLOPs.

This yields test-time capacity scaling with predictable latency: active parameter count keeps on growing with context length, and does not saturate quickly like in MoEs. The dense gating and down-projection preserve contextual mixing, while the STEM path supplies additional capacity at low overhead, supporting long-context tasks (multi-document RAG, CoT) with near-constant per-token compute. 1b illustrates how STEM outperforms the dense baseline at longer context lengths. Additional long-context evaluation on LongBench are provided in Appendix A.2.

Table 2: Training hyperparameters by setting. Common: weight decay $= 0.1$, $\beta_1 = 0.9$, $\beta_2 = 0.95$, LR warmup ratio $= 0.01$. Minimum LR is $0.1\times$ peak LR. For 1B pretraining, we follow the OLMO schedule for 5T tokens but stop early at 1T.

| Configuration | 350M Pretrain | 1B Pretrain | 1B Midtrain | 1B Context-Extend |
|---|---|---|---|---|
| Peak LR | 2e-3 | 4e-4 | 3.2e-4 | 1e-5 |
| LR schedule | cosine | cosine | linear | cosine |
| Batch size | 512 | 512 | 512 | 64 |
| Max sequence length | 2048 | 4096 | 4096 | 32768 |
| Training steps | 100,000 | 500,000 | 50,000 | 10,000 |
| Cross-doc masking | No | No | No | Yes |

## 4 EXPERIMENTS

We evaluate STEM against dense and MoE baselines on downstream tasks while controlling for (i) training compute (activated FLOPs) and (ii) the number of training tokens. MoE variants are configured to match STEM's total parameter count, and their activated FLOPs are kept comparable to the dense baseline. (Note: STEM uses strictly fewer per-token FLOPs than both baselines.) We study two model scales — 350M and 1B, performing comprehensive ablations at 350M and validating STEM at 1B under both pretraining-from-scratch and mid-training insertion. Finally, we assess long-context behavior by further fine-tuning with extended context length. We evaluate the Return on Investment (ROI)—defined here as the ratio of model accuracy to training FLOPs—to determine the training efficiency of each model, as the economic value has become a major concern of foundational models. Formally, we define it as:

$$\text{Training ROI} = \frac{\text{Model Accuracy (Avg)}}{\text{Total Training FLOPs}}$$

### 4.1 EXPERIMENTAL SETTING

**Datasets.** For pretraining, we use OLMO-MIX-1124 (OLMo et al., 2025), a 3.9T-token corpus built from DCLM (Li et al., 2025a) and Dolma 1.7 (Soldaini et al., 2024); we subsample 1T tokens for our runs. For mid-training, we mix OLMO-MIX-1124 (65%), NEMOTRON-CC-MATH-V1 (5%) (Rabeeh Karimi Mahabadi, 2025), and NEMOTRON-PRETRAINING-CODE-V1 (30%) (NVIDIA et al., 2025). For context-length extension, we use PROLONG-DATA-64K (Gao et al., 2024) (63% long-context / 37% short-context) and pack sequences up to 32,768 tokens with cross-document attention masking.

**Models.** We use model architectures from `MobileLLM-350M` (Liu et al., 2024) and `Llama3.2-1B` (Meta AI, 2024) for evaluations. In both the models, we do not share the input embeddings and the language model head. Unless otherwise noted, one third of FFN layers are replaced at uniform intervals by the sparse alternative. For STEM, the dense up-projection is replaced by an embedding table of size $V \times d_{\text{ff}}$ in each layer. For Hash layer MoE design, we use top-1 routing and choose the number of experts per layer to match STEM's total parameter count, while keeping activated FLOPs comparable to the dense baseline. We also report ablations that replace one half of FFN layers with STEM, and an extreme setting that replaces all FFN layers except the first.

**Evaluations.** Pretrained checkpoints are evaluated zero-shot on eight common-sense reasoning tasks: ARC-Easy, ARC-Challenge (Clark et al., 2018), BoolQ (Clark et al., 2019), PIQA (Bisk et al., 2020), SIQA (Sap et al., 2019), HellaSwag (Zellers et al., 2019), OpenBookQA (Mihaylov et al., 2018), and WinoGrande (ai2, 2019). To assess advanced knowledge and mathematical reasoning for mid-training checkpoints, we report MMLU (Hendrycks et al., 2021) and GSM8K (Cobbe et al., 2021). For long-context behavior after context extension, we use Needle-in-a-Haystack (NIAH) (Kamradt, 2024).

**Training details.** We pretrain the 350M models on 100B tokens and the 1B models on 1T tokens. We use the AdamW optimizer with a cosine learning rate schedule, 10% warmup, and a minimum

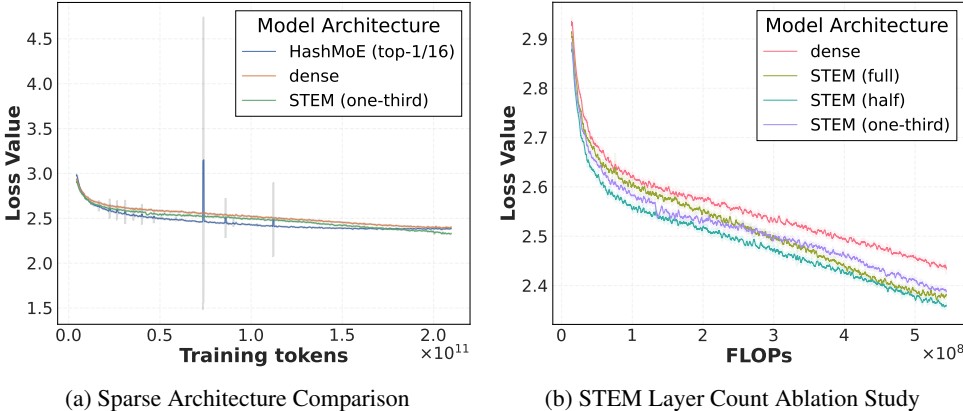

(a) Sparse Architecture Comparison    (b) STEM Layer Count Ablation Study

Figure 3: (a) **Training Stability.** Unlike Hash layer MoE, the 350M STEM model does not show any training loss spikes. (b) **Performance scaling with more STEM layers.** With more STEM layers, a lower training loss can be achieved at fewer training FLOPs.

learning rate of $0.1\times$ the peak value. After pretraining, we run a midtraining stage on 100B tokens, followed by a context-extension stage on 20B tokens. The full set of hyperparameters is listed in Table 2.

## 4.2 EXPERIMENTAL RESULTS

STEM demonstrates the benefits of fine-grained sparse scaling by improving downstream performance with fewer training FLOPs. Interestingly, STEM does not suffer from training instability issues that is often the case for fine-grained MoE models (Databricks, 2024; Dai et al., 2024). Instead, the geometric properties of the STEM embedding spaces further help improve the training convergence. Figure 3a demonstrates the training stability of STEM compared to token-indexed Hash Layers MoE, where HashMoE has more bumpy jumps during the training. Moreover, we see the STEM architecture has larger model capacity (lower training loss tendency) when we scale up the training tokens as the loss curve of STEM crosses over the other two architectures when training tokens increase. Furthermore, even with fewer training FLOPs STEM achieves lower training 3b and validation 1a losses.

## 4.3 DOWNSTREAM EVALUATION RESULTS

We compare STEM with dense baseline as well as Hash layer MoE at 350M scale. On the other hand, for 1B model, we compare STEM (with one-third of FFN replacement) with only the dense baseline. In both cases 3, we observe substantial improvement in tasks requiring comparatively more external knowledge such as, Arc-Challenge and OpenBookQA, while having modest improvements on the rest of the tasks. Additionally, the improvements on the knowledge-intensive tasks are more significant with increase in FFN replacement with STEM layers. Note all the STEM replacement are replacing the up-projection component of original FFN unless specified in the table.

Upon midtraining 4, the 1B STEM model continues to outperform the dense baseline on the language modeling downstream tasks. Additionally, STEM architecture exhibits improvements in reasoning and knowledge retrieval abilities through GSM8k and MMLU performances.

## 4.4 ABLATION STUDIES

### 4.4.1 IMPACT OF STEM LAYER COUNT

To identify the efficacy of STEM layers, we vary the number of FFN layers we replace with STEM alternative. We place the STEM-based decoder layers at regular intervals, interleaved with regular

---

[1]ROI is normalized at each basline for better comparison.

[2]STEM defaults to replacing one third of FFN layers, also writes as STEM-1/3

Table 3: Downstream accuracy of pretrained models at 350M and 1B scales. We report the total number of parameters and the number of active parameters for each model variant. Baseline denotes the dense SwiGLU FFN model. For 350M, in the first few rows, we compare sparse alternatives under similar FLOPs: Hash-MoE (top-1/16 experts in 1/3 of FFN layers), STEM with 1/3 of FFN layers replaced (including up projection replacement, gate projection replacement, and STEM[†] with an additional up-projection). In the next set of rows, we compare STEM with varying up projection layer replacement ratios (1/3, 1/2, full). For 1B, we report the dense baseline and STEM with 1/3 up projection layer replacement.

| Model | #Total Params (B) | #Active Params (B) | ARC-E | ARC-C | BoolQ | PIQA | SIQA | HSwag | OBQA | Wino | Avg | #GFLOPs | ROI[1] |
|---|---|---|---|---|---|---|---|---|---|---|---|---|---|
| | | | | | *350M (Pretraining)* | | | | | | | | |
| Baseline | 0.37 | 0.37 | 57.66 | 30.55 | 58.20 | 69.42 | 41.10 | 49.68 | 34.80 | 56.35 | 49.72 | 0.74 | 1x |
| Hash-MoE | 1.22 | 0.37 | 58.88 | 36.33 | 55.44 | 70.21 | 43.55 | 47.56 | 39.26 | 53.44 | 50.58 | 0.74 | 1.02x |
| STEM [2] | 1.14 | 0.35 | 63.01 | 32.68 | 60.31 | 70.18 | 39.76 | 52.38 | 33.00 | 55.88 | 50.90 | 0.70 | 1.08x |
| STEM (gate-proj) | 1.14 | 0.35 | 54.56 | 34.12 | 59.13 | 64.92 | 44.56 | 43.62 | 36.91 | 55.00 | 49.10 | 0.70 | 1.04x |
| STEM[†] | 1.21 | 0.35 | 57.94 | 34.45 | 59.10 | 68.85 | 43.70 | 45.75 | 41.02 | 53.98 | 50.60 | 0.74 | 1.02x |
| STEM-1/2 | 1.85 | 0.34 | 62.95 | 40.00 | 62.02 | 70.94 | 43.70 | 51.49 | 46.68 | 55.78 | 54.20 | 0.67 | 1.20x |
| STEM-full | 3.25 | 0.30 | 62.21 | 39.61 | 61.99 | 70.73 | 43.60 | 48.44 | 44.53 | 56.33 | 53.43 | 0.60 | 1.33x |
| | | | | | *1B (Pretraining)* | | | | | | | | |
| Baseline | 1.50 | 1.50 | 66.98 | 41.88 | 64.21 | 73.44 | 44.09 | 59.65 | 39.84 | 56.48 | 55.82 | 3.00 | 1x |
| STEM | 6.75 | 1.41 | 65.95 | 42.03 | 61.66 | 75.00 | 44.78 | 60.37 | 45.90 | 57.34 | 56.63 | 2.83 | 1.08x |

Table 4: Mid-trained model evaluations (1B).

| Model | ARC-E | ARC-C | BoolQ | PIQA | SIQA | HellaSwag | OBQA | Winogrande | Avg | GSM8K | MMLU |
|---|---|---|---|---|---|---|---|---|---|---|---|
| | | | | | *1B (Mid-training)* | | | | | | |
| Baseline | 70.78 | 42.11 | 65.84 | 72.95 | 47.13 | 60.39 | 42.97 | 57.81 | 57.50 | 44.2 | 29.92 |
| STEM | 69.78 | 44.22 | 68.54 | 74.69 | 45.65 | 61.90 | 45.70 | 57.42 | 58.49 | 46.4 | 32.38 |

FFN-based decoder blocks. Table 3 shows that increasing the number of replacement from one-third to half improves the average downstream performance substantially. However, the improvement slows down beyond that. Note that, with increasing number of replacements, the training FLOPs also decrease, and therefore the overall training ROI still increases. We can see that the STEM (STEM-1/3) achieves 1.08x training ROI of the baseline, while STEM-1/2 achieves 1.20x and STEM-full achieves 1.33x of the baseline. Figure 3b presents the comparison of the three variants in terms of loss vs training FLOPs.

### 4.4.2 IMPACT OF STEM PLACEMENT

Placement of STEM inside the gated FFN matters. To demonstrate this, we compare two options: replacing the *up-projection* vs. the *gate-projection*. As shown in Table 3, replacing the gate underperforms even the dense baseline, while replacing the up-projection yields consistent gains. In SwiGLU, the gate $\sigma(W^g x)$ should depend on the current hidden state $x$ to modulate $\phi(W^u x)$ contextually. Swapping $W^g x$ for a token-indexed embedding $e_t$ makes the gate largely input-independent ($\sigma(e_t)$), weakening its context-aware selection. Moreover, the nonlinearity can be effectively abstracted away by the learned embeddings, and consequently its role is weakened. In contrast, applying STEM to the up-projection preserves contextual information in gate computation path and proves to be an optimal fine-grained sparse design.

### 4.4.3 UP-PROJECTION WITH ADDITIVE EMBEDDING

To further study the optimality of STEM's design, we implement STEM[†] 2.3, that retains up projection and additively modulates its output with the STEM embedding. Although it adds more parameters and FLOPs, the downstream performance does not improve.

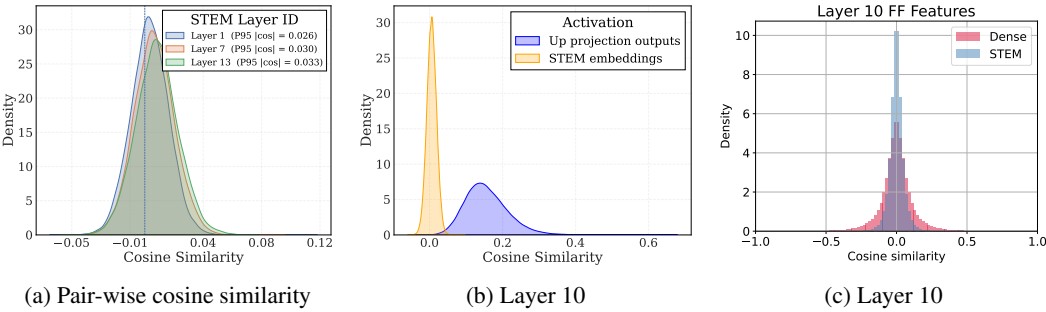

(a) Pair-wise cosine similarity  (b) Layer 10  (c) Layer 10

Figure 4: **Geometry of STEM embeddings.** (a) Distribution of pairwise cosine similarity of STEM embeddings of sampled layers. (b) Pair-wise cosine similarity distributions of up-projection output space and STEM embeddings. (c) Cosine similarities are computed between the input hidden states of the down projection matrix. All the plots are provided from the 1B model.

## 5 STEM CHARACTERISTICS

In this section, we analyze some of the characteristics that STEM embeddings demonstrate. We observe that in each layer the STEM embeddings of different tokens have very low pairwise cosine similarity which elicits some desirable properties regarding information storage capacity and training convergence. Additionally, because of the clear mapping between the embeddings and the tokens, STEM models are more interpretable.

### 5.1 LARGE ANGULAR SPREAD OF STEM EMBEDDINGS

Figure 4a shows that STEM embeddings exhibit very low pairwise cosine similarity—i.e., a large angular spread. We hypothesize that this property improves the information–retrieval behavior of FFN layers by reducing interference among stored items. Prior work (Geva et al., 2021; Meng et al., 2022) models FFNs as key–value memories: each hidden unit is associated with a *key* given by a *row* of the up-projection $W^{(u)} \in \mathbb{R}^{d_{ff} \times d_{model}}$ and a *value* given by the corresponding *column* of the down-projection $W^{(d)} \in \mathbb{R}^{d_{model} \times d_{ff}}$; the gate projection provides context-dependent, multiplicative modulation that creates a selective read. In this view, the pre-activation $h = \phi(W^{(u)}x)$ induces a soft address over memory slots (hidden units).

In contrast, STEM replaces the learned affine addressing with a direct, token-indexed address vector, upon which the gate still applies context-dependent modulation. To quantify the geometry of these address vectors, we report the distribution of pairwise cosine similarities between unit-normalized vectors. A distribution concentrated near zero (as in Figure 4a and Figure 4b) indicates that most angles are close to 90° and thus the angular spread between the vectors is reasonably large. This large angular spread lowers cross-talk between slots and can thereby improve the effective information storage capacity of the FFN memory at fixed width Donoho & Elad (2003); Tropp (2004). Figure 4c demonstrates the distribution of pairwise cosine similarities between the address vectors after the modulation applied by the gate projection.

### 5.2 INTERPRETABILITY OF STEM MODELS

STEM exposes token-indexed, layer-local parameters that act as interpretable FFN *addresses*, enabling simple, reversible edits that causally steer factual predictions with high reliability and low collateral change. Because each token $t$ has a layer-specific STEM vector $e_{t,\ell} \in \mathbb{R}^{d_{ff}}$, we can intervene at inference time in a transparent way.

For example, Figure 5 shows that we can manually control the top next-token probabilities by performing a *swap* at layer $\ell$,

$$e_{\text{Spain},\ell} \leftarrow e_{\text{Germany},\ell},$$

while leaving all other parameters unchanged. Under the original prompt containing "Spain", the intervened model's top-$k$ next-token distribution closely matches that of the control prompt containing "Germany", illustrating precise, token-indexed knowledge editing.

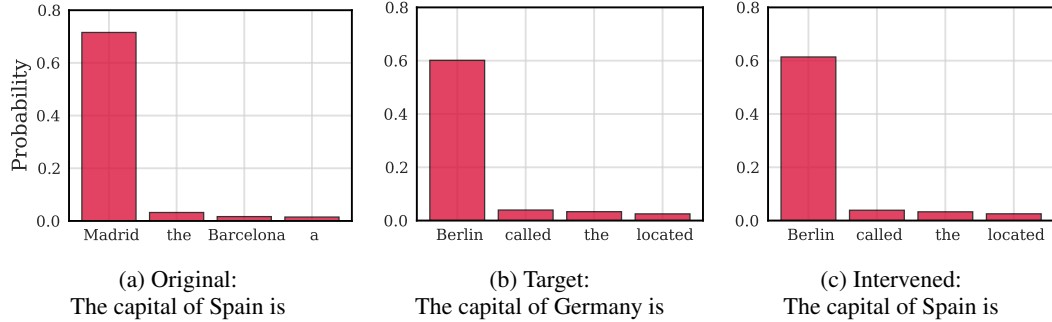

(a) Original:
The capital of Spain is

(b) Target:
The capital of Germany is

(c) Intervened:
The capital of Spain is

Figure 5: **Knowledge edit.** Top-4 next-token probabilities for the original prompt *"The capital of Spain is"* (left), for the target prompt *"The capital of Germany is"* (middle), and for the intervened model where we swap the STEM vector $e_{\text{Spain},\ell}$ with $e_{\text{Germany},\ell}$ at every STEM layer keeping the original prompt the same(right). The swap shifts mass from `Madrid` to `Berlin`, demonstrating token-indexed, layer-local, and reversible control of factual predictions.

## 6  RELATED WORKS

MoE (Shazeer et al., 2017; Fedus et al., 2022) introduced large parametric capacity for LLMs at near-constant FLOPs through sparse computation. The success of MoE models hinges closely with auxiliary loss function designs (Fedus et al., 2022; Rajbhandari et al., 2022; Qiu et al., 2025), and system-level solutions (Huang et al., 2024; Go & Mahajan, 2025; Wang et al., 2024b) that ensure load balance among expert networks, training stability, mitigation of representation collapse (Chi et al., 2022), and tolerable communication overload during training and inference. To avoid the interference of auxiliary routing losses with the training objective, recent works have proposed auxiliary loss-free approaches (Roller et al., 2021; Wang et al., 2024a) that inject fixed or dynamic routing bias to the MoE model.

Conversely, PKM models (Lample et al., 2019) reserve a large key-value parametric memory with efficient top-k selection through memory-efficient keys arranged in product space. PKM(Lample et al., 2019; He, 2024) scales up the parametric memory compared to MoE, increases the granularity of sparsity, and avoids the cross-device communication overhead, but at the cost of high memory lookup cost during inference, and under-training issues of the large value memory. These challenges require sophisticated architectural modifications (Huang et al., 2025) and advanced system-level solutions (Berges et al., 2024) to be overcome.

Recently, Gemma-3n (Google DeepMind, 2024) proposed Per Layer Embeddings (PLE) for small on-device models to *complement* their limited parametric capacity with token-indexed sparse parametric memory. However, they do not dispose of original FFN modules, and use a much lower-dimensional PLE only to modulate the FFN output in each layer. These embedding tables are accommodated in fast storage, outside GPU HBM memory to accommodate larger batch sizes and enable fast prefetching.

## 7  CONCLUSION

This work introduced STEM, a static, token-indexed design that replaces the FFN up-projection with a layer-local embedding lookup.This decouples parametric capacity from per-token compute and cross-device communication, yielding lower per-token FLOPs and fewer parameter accesses, and enabling CPU offload with asynchronous prefetch. Empirically, STEM trains stably despite extreme sparsity (compared to fine-grained MoE variants), improves accuracy over dense baselines, and exhibits higher effective memory capacity via a large-angular-spread embedding space. It also strengthens long-context performance by activating more distinct parameters as sequence length grows, providing practical test-time capacity scaling.

## 8 ETHICS STATEMENT

This work develops and empirically evaluates a novel large language model architecture. All training and evaluation datasets are publicly available and widely used within the research community; no new human-subject data were collected, and no sensitive or proprietary data sources were used. Due to computational resource constraints, experiments were conducted on models with up to one billion parameters and evaluated at pre-training and mid-training checkpoints, and the scope of the conclusions should be interpreted accordingly. Future research in this direction should continue to assess ethical considerations throughout model development, evaluation, and potential deployment.

## 9 REPRODUCIBILITY STATEMENT

This work follows the reproducibility recommendations of ICLR; details necessary to replicate results are referenced rather than repeated here. Section 4.1 documents the training datasets, model architectures, training procedures, and evaluation datasets and protocols referenced throughout the experiments. To facilitate independent verification, code and trained model checkpoints will be released to support full reproducibility.

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

## A  APPENDIX

### A.1  ADDITIONAL BENCHMARKS

Interestingly, our additional experiments on more contextual reasoning-heavy tasks show that STEM's contextual reasoning skill is better than that of the dense baseline. To directly probe reasoning beyond parametric knowledge, we evaluate 1B-scale baseline and STEM models on *BIG-Bench Hard* (BBH) (Suzgun et al., 2022), *MuSR*(Sprague et al., 2024), and the *LongBench*(Bai et al., 2024) multi-hop reasoning and code-understanding subsets. BBH is a collection of diverse, challenging tasks designed to require multi-step and compositional reasoning. MuSR requires the model to track entities and constraints over a long narrative before answering a question. The LongBench multi-hop subset tests reasoning across multiple passages, while the code-understanding subset evaluates comprehension of complex code snippets. As shown in Table 5, STEM consistently outperforms the dense baseline on BBH, MuSR, and on LongBench multi-hop and code-understanding tasks across all context-length ranges, indicating that STEM does not impair contextual reasoning and can in fact improve it.

Table 5: Contextual reasoning benchmarks for 1B-scale models. LongBench scores are averaged over tasks within each context-length range.

| Model | BBH | MuSR | LongBench Multi-hop | | | LongBench Code | | |
|---|---|---|---|---|---|---|---|---|
| | | | $< 4k$ | $4$–$8k$ | $\geq 8k$ | $< 4k$ | $4$–$8k$ | $\geq 8k$ |
| Baseline | 24.87 | 35.85 | 5.72 | 6.20 | 6.19 | 45.37 | 44.64 | 41.30 |
| STEM | 27.55 | 36.38 | 10.20 | 8.63 | 7.82 | 52.68 | 52.53 | 49.60 |

### A.2  ADDITIONAL LONG-CONTEXT EVALUATION

Apart from the synthetic task Needle-in-a-haystack, we further evaluate STEM on LongBench, a long-context benchmark that spans six task categories, including single- and multi-document question answering, summarization, few-shot learning, synthetic tasks, and code completion. We group

Table 6: LongBench results (average across tasks) for 1B models, grouped by context length.

| Model | 0–2k | 2–4k | 4–6k | 6–8k | 8–10k | 10–12k | 12k+ |
|-------|------|------|------|------|-------|--------|------|
| Base  | 24.0 | 23.8 | 22.1 | 22.3 | 21.9  | 21.1   | 23.5 |
| STEM  | 27.6 | 27.6 | 24.4 | 22.7 | 23.0  | 21.7   | 24.2 |

test examples by context length and report the average scores in each regime. As shown in Table 6, the 1B STEM model consistently matches or outperforms the 1B dense baseline across all context-length ranges, indicating that its long-context capabilities extend beyond synthetic tasks.

