# OpenReview forum: "STEM: SCALING TRANSFORMERS WITH EMBEDDING MODULES"
_ICLR.cc/2026/Conference — ICLR 2026 Poster_

### Official Review · Reviewer_eAWT · 2025-10-19

**Soundness:** 2
**Presentation:** 3
**Contribution:** 2
**Rating:** 2
**Confidence:** 2

**Summary:**

This paper proposes STEM, a sparse transformer architecture that replaces the FFN up-projection with token-indexed embeddings from a layer-local lookup table. The authors claim improved training stability, better downstream performance, and enhanced long-context capabilities compared to dense baselines and MoE alternatives.

**Strengths:**

1. The paper identifies real issues with fine-grained MoE models (training instability, load balancing, communication overhead) and proposes a simpler alternative.

2. The paper provides evaluation across multiple scales (350M, 1B), training regimes (pretraining, midtraining, context extension), and benchmarks, as well as systematic ablations on layer count, placement (gate vs. up-projection), and hybrid variants.

3. Analysis of embedding space properties (angular spread) is valuable.

**Weaknesses:**

1. There is a parameter mismatch in the experiments, in particular, STEM uses 1.14B parameters vs. 0.37B dense baseline at 350M scale.

2. The results are incocsistent, as table 3 shows STEM often underperforms baseline: 350M: ARC-C (32.68 vs 30.55), SIQA (39.76 vs 41.10), 1B: BoolQ (61.66 vs 64.21), PIQA (75.00 vs 73.44 favors STEM but margin is small). Abstract claims "up to 3-4% improvement" but average gains are ~1% or less.

3.  The long-context analysis is limited, as NIAH is a synthetic task.

**Questions:**

1. Can you provide results with parameter-matched dense baselines?

2. What are actual wall-clock training/inference times and memory usage?

3. How does this compare to DeepSeekMoE, Mixtral, or other modern MoE methods?

---

> ### Author Response · Authors · 2025-11-23
> **Response to Reviewer eAWT (part 1/3)**
>
> We appreciate the reviewer's suggestions and questions. We are glad that the reviewer appreciates the exhaustivity of our experiments, and the explanation of STEM characteristics. Below we provide clarifications to questions regarding parameter mismatch **(W1, Q1)**, performance improvements **(W2)**, soundness of long-context analysis **(W3)**, comparison with MoE architectures **(Q3)**. In addition, we provide wall-clock training and inference times to validate the efficiency of STEM. We hope our detailed clarification with further experiment results will address the doubts about the soundness of our work.
>
> ---
>
> ### **(W1, Q1) There is a parameter mismatch in the experiments, in particular, STEM uses 1.14B parameters vs. 0.37B dense baseline at 350M scale. Can you provide results with parameter-matched dense baselines?**
>
> We thank the reviewer for raising this clarification question. In our experiments, we match models by the number of **active parameters per token** and the corresponding **FLOPs per token**, rather than by total parameter count. This follows common practice in the sparse-model literature, where the goal is to increase parametric capacity *without* increasing training or inference-time compute or memory usage.
>
> Under this active-parameter view, STEM is actually *more efficient* than the dense baseline. The table below illustrates the comparison for the 350M setting (forward-only FLOPs for sequence length \(L = 2048\)):
>
> | Model                                | Total Params (B) | Active Params / Token (B) | Training FLOPs / Token (G, forward, L=2048) |
> |--------------------------------------|------------------|---------------------------|---------------------------------------------|
> | Dense 350M baseline                  | 0.37             | 0.37                      | 2.82                                        |
> | STEM (replace 1/3 up-proj)           | 0.87             | 0.35                      | 2.67                                        |
> | STEM (replace 1/2 up-proj)           | 1.14             | 0.34                      | 2.61                                        |
> | STEM (replace all but 1 up-proj)     | 1.87             | 0.30                      | 2.40                                        |
> | Hash-MoE (16 experts, 1 active, 1/3 layers) | 1.22       | 0.37                      | 2.82                                        |
>
> As shown, even though STEM variants have a larger **total** parameter count, their **active** parameters and FLOPs per token are *strictly lower* than the dense baseline. A “parameter-matched” dense baseline in terms of **total** parameters (e.g., a 1.14B dense model) would require substantially higher FLOPs and memory per token, giving it an unfair advantage under a fixed compute and memory budget.
>
> To provide a more meaningful comparison at matched active parameter counts, we also include a Hash-MoE variant where 1/3 of the layers are MoE layers. This Hash-MoE model matches the dense baseline in active parameters per token and has a slightly larger total parameter count than the STEM model with 1/3 of the MLP layers replaced by STEM layers, offering a fairer comparison under equivalent computational constraints.
>
> ---
>
> ### **(W2) Abstract claims "up to 3-4% improvement" but average gains are ~1% or less. Inconsistent improvements on different benchmarks.**
>
> We thank the reviewer for pointing this out. The phrase “up to 3–4% improvement” in the abstract refers to the largest average gain we observe across our settings. In particular, for the 350M model with half of the up-projection layers replaced by STEM, we see more than a **4-point absolute improvement** (54.20 - 49.72) in average downstream accuracy under **lower training FLOPs**:
>
>
> | Model                        | ARC-E | ARC-C | BoolQ | PIQA  | SIQA | HSwag | OBQA  | Wino  | Avg   |
> |------------------------------|-------|-------|-------|-------|------|-------|-------|-------|-------|
> | Baseline                     |  57.66 | 30.55 | 58.20 | 69.42 | 41.10| 49.68 | 34.80 | 56.35 | 49.72 |
> | STEM (replace 1/2 up-proj)          |  62.95 | 40.00 | 62.02 | 70.94 | 43.70| 51.49 | 46.68 | 55.78 | 54.20 |
>
> As the reviewer notes, at other scales and on some benchmark suites the average gains are smaller (around 1 point), and performance can vary across individual tasks. This is common when comparing different architectures over a diverse task suite. Following prior work [1,2], we therefore focus on the average score as the primary summary metric, which captures overall performance across heterogeneous downstream tasks.
>
> To avoid any ambiguity, we have revised the abstract to explicitly state that the 3–4% figure refers to the maximum average improvement observed and will mention that typical gains are around 1–4 points depending on the scale and benchmark.
>
> [1] OLMo: Accelerating the Science of Language Models (Groeneveld et al., 2024)
>
> [2] 2 OLMo 2 Furious (Team OLMO et al., 2025)

---

> ### Author Response · Authors · 2025-11-23
> **Response to Reviewer eAWT (part 2/3)**
>
> ### **(W3) The long-context analysis is limited, as NIAH is a synthetic task.**
> We thank the reviewer for requesting for more evaluations on the long-context ability of STEM. We further evaluated STEM on the **Longbench** benchmark, which is a long-context evaluation benchmark that spans across six key task categories such as single-document and multi-document question answering, summarization, few-shot learning, synthetic tasks, and code completion. We divide the test examples into different context length regimes. We observe that across regimes the performance of our 1B STEM model is superior to that of the 1B dense baseline model.
> | Model      | 0–2k | 2–4k | 4–6k | 6–8k | 8–10k | 10–12k | 12k+ |
> |-----------|:----:|:----:|:----:|:----:|:-----:|:------:|:----:|
> | Base      | 24.0 | 23.8 | 22.1 | 22.3 | 21.9  | 21.1   | 23.5 |
> | STEM      | 27.6 | 27.6 | 24.4 | 22.7 | 23.0  | 21.7   | 24.2 |
>
> [1] LongBench: A Bilingual, Multitask Benchmark for Long Context Understanding (Bai et al, 2024)
>
> ---
> ### **(Q2) What are actual wall-clock training/inference times and memory usage?**
> We thank the reviewer for raising this question. We provide the throughput improvements and memory savings with STEM below.
>
> **Throughput improvement:** Using our current implementation, we achieve **1.27x speedup in the MLP layers** which is close to the theoretical limit. In the table below, we provide the end-to-end speedups at 350M model scale using an NVIDIA H200 GPU (143.8 GiB HBM3e, 4.8 TB/s memory bandwidth, 600 W TDP).
>
> | Model | MLP Speedup | Training throughput (toks/s) | Prefilling Throughput (toks/s) |Decoding Throughput (toks/s) | Avg Downstream Accuracy (\%) |
> |----|------|------|----|-----|-----|
> |Baseline| - | 128K| 409K | 70.5K | 49.72 |
> |STEM    | 1.27x | 145K (1.13x) | 471K (1.15x) | 80.7K (1.14x) | 53.43 |
>
> > Note that at a 1.13x training speedup, we achieve average downstream task performance improvement of 3.6\% at 350M model scale.
>
> Our current implementation gives a 1.13x improvement in training throughput, which translates to roughly 12\% savings in GPU hours. In the paper, we have elaborated on the theoretical speedup that can be expected (Sec 3.1, Page 4,5). As a follow-up, we shall work on improving our current implementation to match the expected speedups.
>
> **HBM Usage:** We only need to prefetch the embeddings with respect to the **unique tokens**. In practice, for realistic batch sizes, the number of unique tokens per step is much smaller than the total number of batched tokens. As a result, the memory needed for prefetched STEM embeddings is much smaller than that for activations and gate outputs, while we also eliminate the HBM footprint of the up-projection weights and their optimizer states. For additional speedup during decoding, we maintain an LFU cache of most frequently prefetched embeddings. This LFU cache size is strictly maintained to be smaller than the up-projection parameters.
>
> **Implementation details:** We keep the offloaded embedding table onto a *pinned* CPU memory. To improve the prefetching throughput, we implement **kernel-assisted copy**, that lets a CUDA kernel to fetch the required data from CPU with larger number of threads. This is especially beneficial when the retrieved data is scattered in the CPU memory. In addition, we use torch.compile to minimize CPU-side kernel launching overhead.

---

> ### Author Response · Authors · 2025-11-23
> **Response to Reviewer eAWT (part 3/3)**
>
> ### **(Q3) How does this compare to DeepSeekMoE, Mixtral, or other modern MoE methods?**
>
> We thank the reviewer for this question. Our experiments are conducted at relatively small scales (350M and 1B), where we found that **fine-grained MoE models are difficult to train stably** without substantial additional engineering. Modern systems such as Mixtral, DBRX, and DeepSeekMoE operate at much larger scales and rely on a combination of carefully designed auxiliary losses, routing/balancing regularizers, and system-level optimizations to achieve strong performance [1,2]. Replicating those setups faithfully is beyond the scope of our current work and resource budget, so we do **not** claim a direct, apples-to-apples comparison with those specific 7B–100B+ MoE models.
>
> Instead, at the 350M/1B scales we target, we compare STEM against a **Hash-MoE baseline** with static expert selection, which is significantly more stable than fine-grained routed MoEs in our setting. This Hash-MoE baseline matches the dense model in active parameters per token and FLOPs, and has slightly more total parameters than our STEM configuration with 1/3 of the MLP layers replaced by STEM, providing a fair comparison under matched compute constraints. Across these settings, STEM achieves better average downstream accuracy while also reducing per-token FLOPs.
>
> We want to highlight that We view STEM as a **complementary architectural primitive** rather than a replacement of MoE. STEM improves the design of the *per-layer gated FFN* by replacing the up-projection with a token-indexed embedding module. In principle, this idea can be combined with MoE routing—e.g., by designing MoE layers whose experts are themselves STEM-FFNs instead of standard dense FFNs. Exploring such “Mixture-of-STEM-Experts” variants at larger scales is an exciting direction for future work.
>
> [1] DBRX (Mosaic Research Team, Databricks, 2024)
> [2] DeepSeekMoE: Towards Ultimate Expert Specialization in Mixture-of-Experts Language Models (Dai et al., 2024)

---

### Official Review · Reviewer_7Z8s · 2025-10-30

**Soundness:** 3
**Presentation:** 2
**Contribution:** 3
**Rating:** 6
**Confidence:** 3

**Summary:**

The paper introduces STEM, which is a static, token-indexed design that replaces the FFN up-projection with a layer-local embedding lookup.This decouples parametric capacity from per-token compute and cross-device communication, yielding lower per-token FLOPs and fewer parameter accesses, and enabling CPU offload with asynchronous prefetch. STEM is a static, token-indexed, fine-grained mechanism that replaces only the up-projection in gated FFNs with a token-specific vector retrieved from a layer-local embedding table.

**Strengths:**

STEM simplifies operation compared to MOE by dense up-projection in the SwiGLU FFN with a token-indexed vector from a per-layer table lookup.

STEM improves both computation by reducing per-layer FLOPs during training and memory access by lowering parameter traffic relative to a dense up-projection.

Experiments are informative, with sufficient ablations. The paper does a good job at describing the various aspects of the STEM technique and in pointing out the benefits of the approach relative to other methods such as MOE. Design choices are clearly (if tersely) motivated.

**Weaknesses:**

Paper is dense and focused. It uses a lot of jargon and will not be so accessible to readers not familiar with the ideas and issues specific to this narrow research area. This is not necessarily a weakness, but readers that are not in this area may not appreciate the paper's purpose or contributions. The paper would be better with some diagrams to illustrate the architectural differences between STEM and other approaches, including MOE techniques. Figure 1.c is too small and unclear for this purpose.

In section 4.4.1 the term "ROI" is used in reference to table 3. However, table 3 does not include ROI in the numbers. What does the acronym ROI stand for? "Return on Investment"? That doesn't really make sense in this context (what is the "investment"? What is the "return"?). The text only vaguely defines what is meant by ROI, as "performance over training flops". I assume that "over" in this context means divided by, and performance means the average of the scores. This should be expressed more clearly and precisely, e.g. as an equation "ROI = average accuracy/#training flops".

Table 3 (and other tables in the paper) is lacking an informative caption. There is no definition of the numbers, just the column headings. The column headings just state the problem task (presumably). The caption should say that these numbers represent percent correct on the task test sets.

It is not clear how figure 3 expresses anything about the "Geometry" (which is misspelled here) of the STEM Embeddings. All that it shows are histograms of cosine distances.

**Questions:**

What are the limitations of the STEM method compared to the competing approaches such as MOE? Presumably something is lost due to the increased sparsity compared to MOE.

---

> ### Author Response · Authors · 2025-11-23
> **Response to Reviewer 7Z8s (part 1/2)**
>
> Thank you for providing such constructive feedback. We are glad that you appreciate the simplicity and ingenuity of our architecture design, and the exhaustiveness of our ablation studies. Based on your recommendations, we have provided more illustrative figures of architectural details and have rewritten the technical details with more lucid text **(W1)**. We clarified the ambiguous definitions and table captions **(W2, W3)**. We have tried to provide more explanations for your queries regarding STEM embedding geometry and its limitations. We hope that our responses answer some of your questions, and we would be happy to receive more feedback about the presentation of our updated manuscript.
>
> ---
>
> ### **(W1) Better illustrative figures of architecture details. Lucid explanation of technical details.**
> Thank you for this constructive suggestion. We have provided additional illustrative figures (Figure 2, Page 3) explaining the architectural differences between the dense baseline (gated FFN), MoE, and STEM alternative.
>
> ---
>
> ### **(W2) What does the acronym ROI stand for? The text only vaguely defines what is meant by ROI, as "performance over training flops".**
> We thank the reviewer for pointing out the ambiguity regarding the acronym ROI. In the context of our paper, ROI stands for "Return on Investment."
> We adapt this economic term to measure computational efficiency, where the "investment" is the computational budget (Training FLOPs) and the "return" is the downstream model performance.
> Formally, we define it as:
> $$\text{Training ROI} = \frac{\text{Model Accuracy (Avg)}}{\text{Total Training FLOPs}}$$
>
> This metric allows us to normalize performance against computational cost, highlighting methods that achieve higher accuracy with fewer resources. We have revised the paper to explicitly define this acronym and the associated formula to ensure clarity for future readers (Sec 4, Page 5).
>
> ---
>
> ### **(W3) Uninformative caption for Table 3**
> We thank the reviewer for this constructive suggestion. We have provided a more informative caption for Table 3 (Page 7).
>
> ---
>
> ### **(W4) It is not clear how figure 3 expresses anything about the "Geometry" (which is misspelled here) of the STEM Embeddings. All that it shows are histograms of cosine distances.**
> Thank you for pointing it out. We have corrected the typo in our updated manuscript.
> The goal of figure 3 is to use **pairwise cosine similarity as a simple quantitative proxy for how spread out the address vectors.** A distribution of pairwise cosines that is tightly concentrated near 0 (and has small upper quantiles of $|\cos\theta_{ij}|$) means most angles $\theta_{ij}$ between individual embedding vectors $u_i$ and $u_j$ are close to $90^\circ$. Consequently the vectors are nearly orthogonal to each other and thus have **large angular spread**. In contrast, large positive cosines indicate that many vectors are nearly collinear, corresponding to a **narrow** angular spread.
>
> In the “FFN as key–value memory’’ view [1], this behavior is closely related to *mutual coherence* of the memory addresses defined by up-projection output vectors. Low coherence is known from sparse coding and dictionary learning to reduce interference between stored items and increase effective memory capacity at fixed width [2,3]. Our measurements show that STEM address vectors have systematically lower pairwise cosine similarity than the dense up-projection outputs, indicating a more angularly dispersed, lower-coherence code and therefore less cross-talk between FFN memory slots. We have clarified this connection in the paper (Sec. 5.1, Page 8) and explicitly state that we use the distribution of pairwise cosine similarities as a coarse but informative descriptor of the embedding geometry.
>
> [1] Transformer Feed-Forward Layers Are Key-Value Memories (Geva et al, 2021)
>
> [2] Optimally sparse representation in general (nonorthogonal) dictionaries via $\ell1$ minimization (Donoho et al, 2003)
>
> [3] Greed is Good: Algorithmic Results for Sparse Approximation (Tropp et al, 2004)

---

> ### Author Response · Authors · 2025-11-23
> **Response to Reviewer 7Z8s (part 2/2)**
>
> ### **(Q1) What are the limitations of the STEM method compared to the competing approaches such as MOE? Presumably something is lost due to the increased sparsity compared to MOE.**
>
> We thank the reviewer for this thoughtful question. Conceptually, we see STEM as **complementary** to Mixture-of-Experts (MoE), not a strict replacement. STEM mainly improves the *per-layer gated FFN* by replacing the up-projection with a token-indexed embedding module, and could in principle be combined with MoE routing, e.g., by using STEM-FFNs as the experts in a “Mixture-of-STEM-Experts” layer so that MoE-style conditional computation and STEM’s efficiency gains coexist.
>
> That said, compared to a well-trained MoE, STEM can be **less context-adaptive**: in MoE, expert selection is context-dependent, whereas in STEM the “expert” is effectively chosen by the token identity and then modulated by a gate, which may limit expressivity in regimes where very fine-grained context routing is crucial. Empirically, however, at the 350M and 1B scales we study, we do **not** observe this as a practical limitation. STEM improves average downstream accuracy while reducing per-token FLOPs and avoiding the training instability and system complexity of fine-grained MoE. This makes STEM a practical alternative at these scales and a promising **building block** for future MoE-style architectures.

---

### Official Review · Reviewer_8jLf · 2025-11-01

**Soundness:** 2
**Presentation:** 2
**Contribution:** 3
**Rating:** 6
**Confidence:** 3

**Summary:**

The paper introduces STEM, a new transformer architecture that replaces the FFN up-projection with a token-indexed, layer-local embedding lookup. This approach offers static sparsity, removing runtime routing and enabling CPU offload, while keeping gate and down-projection layers dense. Experiments on 350M and 1B parameter models show consistent improvements compared with dense and Hash-MoE baselines, with no observed instability. Theoretical analysis, ablations, and interpretability experiments (e.g., token embedding swapping) support the design’s efficiency and explainability advantages.

**Strengths:**

1. Novel Architecture Design. A creative and simple static-sparsity alternative to MoE models that avoids routing overhead and load-balancing complications.
2. Strong Empirical Validation. Comprehensive experiments across scales (350M, 1B) and tasks demonstrate consistent benefits in both efficiency and accuracy.
3. Training Stability. Unlike many fine-grained sparse models, STEM avoids loss spikes and under-trained experts.

**Weaknesses:**

1. Potential Memory Overhead. Each layer’s token-indexed embedding table may become impractical for large vocabularies, despite CPU offload?
2. Lack of Ablation on Embedding Dimensionality. The effect of embedding size or vocabulary size on performance and stability is unexplored.

**Questions:**

1. How does STEM scale beyond 1B parameters? Are there any observed communication or memory bottlenecks at larger scales (e.g., 7B+)?
2. What is the exact CPU-GPU data transfer overhead in practical deployments? Are there latency trade-offs during inference?
3. Could we have some analysis on inference and training efficiency?

---

> ### Author Response · Authors · 2025-11-23
> **Response to Reviewer 8jLf (part 1/4)**
>
> Thank you for your constructive feedback. We are glad that you appreciate the novelty of our architecture design, strong empirical validation, and training stability characteristics of STEM. We address your concern about the impracticality of large vocabulary sizes **(W1)**, embedding dimension ablation study **(W2)**, scalability of STEM **(Q1)**, and communication overhead during inference **(Q2)**. We have also updated our manuscript with a more rigorous study of training and inference efficiency of STEM **(Q3)**. We hope that these responses answer some of your questions, and would look forward to any further comments.
>
> ### **(W1) Potential Memory Overhead. Each layer’s token-indexed embedding table may become impractical for large vocabularies, despite CPU offload?**
> Thank you for raising this concern. We observe that despite large vocabulary size, and large embedding dimension of large scale LLMs, the combined STEM embedding table memory footprint is within available limits of some of the existing computing resources available publicly.
>
> To illustrate this, we estimate the size of the STEM embedding table for different model sizes for Qwen3 model series below.
> > NOTE: Qwen3 uses vocabulary size = 151,936 which is a fairly large vocabulary size
>
> | Model        | # Layers L | Intermediate d_ff | 1/3 STEM Replacement (GB)         | 1/2 STEM Replacement (GB)       | Full STEM Replacement (GB)            |
> |--------------|------------|-------------------|-----------------------------------------|----------------------------------------|--------------------------------------------|
> | Qwen3-1.7B   | 28         | 6,144             | 15                    | 24                      | 48                          |
> | Qwen3-4B     | 36         | 9,728             | 33                        | 49                      | 98                          |
> | Qwen3-8B     | 40         | 12,288            | 45                   | 69                     | 140                         |
> | Qwen3-14B    | 40         | 17,408            | 64                   | 98                     | 196                         |
> | Qwen3-32B    | 64         | 25,600            | 152                   |  231                     | 462                         |
>
> We can see that the total CPU memory required for offloading is limited to 500GB, which is easily available in an AWS H200 node, such as *P5e* or *P5en*, which typically has **2TB of CPU RAM** available. Although in case of training, the optimizer states increase this footprint by 3x, the additional memory overhead can still be accommodated.
> Additionally, the embedding memory footprint depends linearly on vocabulary size and embedding dimension. For the same model series, the vocabulary size stays the same and the model memory footprint (apart from the embedding tables) increases almost quadratically with embedding dimension. Thus the rate of memory footprint increase is slower for the STEM embedding tables than the rest of the parameters.
>
> But if we want to further optimize the total memory footprint of the embedding tables, we can look into some additional techniques:
>
> [1] **Quantizing STEM embeddings**: The large angular separation between the learned embeddings can allow us to apply vector quantization techniques to significantly reduce the memory footprint without much performance degradation.
>
> [2] **Better caching strategies**: We can exploit the Zipfian distribution of the tokens to design a better hierarchical memory to store the embeddings. Currently, we have a two level memory hierarchy - a **GPU-resident LFU cache** and a CPU-resident offloaded memory. We can extend this design to the disk storage in the unlikely event of limited CPU memory.

---

> ### Author Response · Authors · 2025-11-23
> **Response to Reviewer 8jLf (part 2/4)**
>
> ### **(W2) Lack of Ablation on Embedding Dimensionality. The effect of embedding size or vocabulary size on performance and stability is unexplored.**
> Thank you for this interesting recommendation. We have worked with two different vocabulary (32000 and 128256) and embedding sizes (2560 and 8192) for the 350M and 1B model scale. We see no stability issues across these two quite different configurations.
>
> If we want to ablate the embedding size for a single architecture, we meet some architectural challenges: the STEM embedding dimension dictates the FFN intermediate size, and thus the entire model architecture. Reducing the STEM embedding size would require us to use a smaller intermediate size for the gate and down projection which can degrade model performance.
>
> However, we do perform an ablation study on a slightly modified architecture that stacks a regular FFN and a STEM FFN layer in each modified decoder block, similar to PLE in Gemma-3n [1]. The STEM FFN intermediate size is given by the STEM embedding size. Conversely, the regular FFN intermediate size is reduced to maintain the *same active parameter count*. We call it *parameter-matched low-dimensional STEM*. We find that this modified architecture underperforms our STEM design.
>
> In our 1B model experiment, we use a reduced STEM embedding size of 1024 and introduce an additional regular FFN with intermediate size of (8192 - 1024) = 7168. Here are the architectual details below.
>
> | Model               | FFN 1                                                                                      | FFN 2                                                                                      |
> |---------------------|--------------------------------------------------------------------------------------------|--------------------------------------------------------------------------------------------|
> | Baseline            | Up projection size: (8192 × 2048), <br> Gate projection size: (8192 × 2048),<br>Down projection size: (2048 × 8192) | —                                                                                          |
> | param-matched low-dim STEM-1K | Up projection size: (7168 × 2048),<br>Gate projection size: (7168 × 2048),<br>Down projection size: (2048 × 7168) | STEM embedding size: 1024,<br>Gate projection size: (1024 × 2048),<br>Down projection size: (2048 × 1024) |
> | STEM                | —                                                                                          | STEM embedding size: 8192,<br>Gate projection size: (8192 × 2048),<br>Down projection size: (2048 × 8192) |
>
> Below, we provide the validation losses vs number of training tokens  during pretraining with OLMO pretraining dataset. For a fair comparison, we modify every third layer of the partial replacement model architecture similar to our existing STEM checkpoint.
>
> | Model                      | 240M  | 320M  | 400M  | 480M  | 560M  | 640M  | 720M  | 800M  | 880M  | 960M  |
> |---------------------------|-------|-------|-------|-------|-------|-------|-------|-------|-------|-------|
> | Baseline       | 13.46 | 13.24 | 13.11 | 12.97 | 12.81 | 12.61 | 12.43 | 12.23 | 12.07 | 11.92 |
> | param-matched low-dim STEM-1K <br> (every 3rd layer)  | 13.14 | 13.00 | 12.80 | 12.63 | 12.51 | 12.35 | 12.24 | 11.99 | 11.74 | 11.54 |
> | STEM (every 3rd layer)  | 12.73 | 12.52 | 12.42 | 12.24 | 12.04 | 11.91 | 11.68 | 11.36 | 10.98 | 10.72 |
>
> At the 1B model scale, we observe that although the modified architecture provides better performance compared to the dense baseline, it still underperforms STEM and have comparatively less training FLOPs savings. Additionally, with reduced STEM embedding size, the inference efficiency is also compromised as the up-projection matrix size increases.
>
> While this study already shows that reducing STEM embedding dimensionality harms performance and efficiency under a fixed parameter budget, a more exhaustive sweep over embedding sizes is an interesting direction for future work.

---

> ### Author Response · Authors · 2025-11-23
> **Response to Reviewer 8jLf (part 3/4)**
>
> ### **(Q1) Scaling beyond 1B? Communication or memory bottlenecks for larger scales?**
> We thank the reviewer for this suggestion. For resource limitations, we have restricted our pretraining experiments to 1B model scale. We wish to study the performance benefits from STEM at larger model scales such as 8B and 32B.
> In our benchmarking, we do not find communication and memory footprint to be a limiting factor for scaling STEM. With increasing model size, the communication and memory overhead is expected to increase linearly with model hidden state size and number of layers, while training resource usage increases quadratically.
>
> We provide the communication latencies and embedding memory footprint (offloaded to CPU) for prefetching STEM embeddings for 2048 unique tokens across all layers for different model sizes below. We consider full STEM replacement as that increases the communication and memory overhead the most. We measure average prefetching latency using an NVIDIA H200 GPU with PCIe5 CPU-GPU interconnect.
>
> | Model        |  Communication latency (ms) | Memory footprint (GB) (x3 to save optimizer states)|
> |--------------|------------|-------------------|
> | Qwen3-1.7B   |       13.92  | 48                          |
> | Qwen3-4B     |       28.24  | 98                          |
> | Qwen3-8B     |       39.61  | 140                         |
> | Qwen3-14B    |       55.45  | 196                         |
> | Qwen3-32B    |      131.63  | 462                         |
>
> We observe that for 1.7B model scale, the prefetching latency is ~10\% of the total training latency, which can further overlapped with computation through asynchronous operation.Thus communication remains a modest fraction of training iteration time at 1.7B model size, and we expect it not to grow faster than compute at larger scales; indeed, the prefetch volume per step is bounded by the number of unique tokens and scales more gently than the total compute.
>
> ---
>
> ### **(Q2) What is the exact CPU-GPU data transfer overhead in practical deployments? Are there latency trade-offs during inference?**
> We thank the reviewer for raising this important question. Because STEM embeddings are token indexed, the embeddings for the later layers can be prefetched during the computation of the earlier layers. This pipelined design along with some additional implementation considerations help us to hide the communication overhead. Prefetching during prefilling is similar to training and comparatively easier as the tokens for the next batch is known beforehand. To ensure decoding efficiency we leverage the following strategies:
>
> [1] **Token deduplication**: In a large batch size, we can expect to have a low fraction of unique tokens. We only require to prefetch the STEM embeddings for the unique tokens and then scatter them to their respective batch positions. This deduplication can significantly save the CPU-GPU transfer bandwidth.
>
> [2] **LFU Cache**: Another important component of our prefetching system is LFU caching of the most frequently appearing token embeddings in a GPU-resident cache. Because of the Zipfian distribution of token frequencies, we can greatly benefit from a large number of cache hits in the local GPU-resident cache.
> We observe the number of cache misses for different batch sizes while generating completions for PG19 sequences with 1B STEM model. We choose the cache size based on the hidden size of the model, such that it matches the memory footprint of the up-projection matrix that is ommitted.
>
> **Implementation details**: STEM allows us to prefetch the STEM embeddings for the later layers asynchronously and concurrently with the computation of the early layers. In order to reduce prefetching latency and maximize throughput, we utilize kernel-assisted copy. Below we provide the average number of cache misses and average prefetching latency using an NVIDIA H200 GPU with PCIe5 CPU-GPU interconnect.
> | Batch size | Avg cache miss per decoding step | Prefetching latency (ms) | Decoding latency (ms) |
> |------------|----------------------------------|--------------------------|------|
> |32 | 7.3    | 0.05 | 2.01|
> |64 | 13.96  | 0.08 | 2.24|
> |128| 27.65  | 0.14 | 2.93|
>
> Thus we can see that in all cases, prefetching accounts for less than ~5% of the per-step decoding latency and is further reduced in practice by overlapping with computation.

---

> ### Author Response · Authors · 2025-11-23
> **Response to Reviewer 8jLf (part 4/4)**
>
> ### (Q3) Could we have some analysis on inference and training efficiency?
>
> We thank the reviewer for asking for a clearer analysis of STEM’s training and inference efficiency. Below we summarize our theoretical comparison to the dense baseline for a *single* decoder layer.
>
> **Training efficiency.**
> Consider a batch of $B$ sequences with sequence length $L$, hidden width $d$, and FFN hidden size $d_{\mathrm{ff}}$. Ignoring elementwise ops and biases, the per-layer training FLOPs (ignoring constant factors for forward + backward computation) are
> $$
> F_{\text{train}}^{\text{base}}
> = B\bigl(4Ld^2 + 2L^2 d + 3 L d \ d_{\mathrm{ff}}\bigr)
> $$
>
> $$
> F_{\text{train}}^{\text{stem}}
> = B\bigl(4Ld^2 + 2L^2 d + 2 L d \ d_{\mathrm{ff}}\bigr)
> $$
>
> The per-layer FLOPs reduction is
> $$
> \Delta F_{\text{train}}
> = F_{\text{train}}^{\text{base}} - F_{\text{train}}^{\text{stem}}
> = B L d\ d_{\mathrm{ff}}
> $$
> with saving fraction
> $$
> \text{saving fraction}
> = \frac{\Delta F_{\text{train}}}{F_{\text{train}}^{\text{base}}}
> = \frac{d_{\mathrm{ff}}}{4d + 2L + 3 d_{\mathrm{ff}}}
> $$
>
> Plugging in the architecture hyperparameters for Qwen2.5 models yields theoretical FFN-layer training savings of $21.7%$ (1.5B), $22.8%$ (3B), $23.9%$ (7B), $19.7%$ (14B), and $24.8%$ (32B).
>
> **Inference efficiency.**
> Prefill has similar compute characteristics to training and thus enjoys the same per-layer FLOPs reduction. Decoding, however, is primarily memory-bound: the dominant cost is loading parameters and KV cache. For batch size $B$ and context length $L$, the per-layer memory access cost is
> $$
> M_{\text{dec}}^{\text{base}}
> = B\bigl(4d^2 + 2L d + 3 d\, d_{\mathrm{ff}}\bigr)
> $$
> $$
> M_{\text{dec}}^{\text{stem}}
> = B\bigl(2L d + 4d^2 + 2 d\ d_{\mathrm{ff}}\bigr)
> $$
> The reduction in parameter loading cost is
> $$
> \Delta M_{\text{dec}}
> = M_{\text{dec}}^{\text{base}} - M_{\text{dec}}^{\text{stem}}
> = B d\ d_{\mathrm{ff}}
> $$
> with saving fraction
> $$
> \text{saving fraction}
> = \frac{\Delta M_{\text{dec}}}{M_{\text{dec}}^{\text{base}}}
> = \frac{d_{\mathrm{ff}}}{4d + 2L + 3 d_{\mathrm{ff}}}
> $$
> which matches the FLOPs saving factor during training and prefill. As batch size increases and the linear layers become more compute-bound, this per-layer reduction in FLOPs ensures that STEM’s decoding efficiency gains are sustained even in the high-throughput regime.
> We have added this analysis in detail in the updated manuscript (Sec 3.1, Page 4).

---

### Official Review · Reviewer_YuJd · 2025-11-01

**Soundness:** 3
**Presentation:** 4
**Contribution:** 3
**Rating:** 8
**Confidence:** 4

**Summary:**

The paper proposes a new modification to the Transformer architecture to enhance fine-grained sparsity. The most popular way to achieve this today is via MoE architectures which are known to the tricky to train
in a stable manner and introduce significant communication overheads during training and inference. In addition MoE training also has to deal with issues such as experts getting under-trained. Load-balancing objectives are commonly used to alleviate this but they come at a certain quality cost typically.
STEM proposes instead looking at the standard SwiGLU FFN architecture and replacing the up-projection layer with a token indexed embedding lookup that is specific to each layer. That is a depth L model will have L+1 embedding tables in total. This is a way of introducing static fine-grained sparsity into deeper layers of the model. The benefits of such a change include minimizing communication overhead, enabling CPU offload with asynchronous prefetch while achieving fine-grained sparsity.

The authors perform pre-training experiments on a 350M parameter model for 100B tokens and a 1B parameter model for 1T tokens. They also perform mid-training and context extension experiments on a 1B model with 100B and 20B token budgets respectively.

The authors observe better training stability with STEM training. They see consistently improved performance on knowledge-intensive tasks.
They also see improved long-context inference performance as STEM activates more distinct parameters as the sequence length grows. STEM also is more efficient in terms of train time and test time FLOPs.
A key concern with approaches like STEM is the gain in shallow parametric capacity comes with losses in deep contextual reasoning abilities. However, the experiments of the paper show some initial promising results to potentially mitigate this concern.

**Strengths:**

- The paper proposes an interesting idea to complement the transformer architecture. The authors are aware of the many practical challenges surrounding today’s transformer architecture and take them into account when designing STEM.

- The proposed method can be applied in stages by deciding to only apply it to some portion of the layers of a transformer. This offers for a smoother transition away from the standard architecture and for greater flexibility in architecture design.

- The paper performs an analysis of the pairwise cosine similarity of STEM embeddings vs original FFN up projection’s embeddings and finds that there is less redundancy in STEM embeddings. While this is an interesting finding, it is unclear whether reduced redundancy might negatively affect robustness.

**Weaknesses:**

- Although the paper correctly accounts for reduced FLOPs, it can often be quite challenging to realize the full benefit of a FLOP reduction in practice. The paper offers a theoretical discussion of how the embedding table reads can be pre-fetched and offloaded onto CPU which makes sense but it would have made a stronger case if wall-clock training, prefill and generation time comaprisons were provided along with HBM usage.

- The concern about the hit to contextual reasoning abilities (since we replace processing of later layers’ highly enriched residual streams with a token indexed embedding) isn’t fully alleviated by the downstream numbers shown. It is nice that we see noticeable improvements in MMLU and GSM8k which are more reasoning heavy tasks but given the scale of the model, the improvement could be coming from the greater parametric knowledge capacity provided by STEM.

**Questions:**

- Can you provide some details on what the wall-clock time or GPU hour impact of STEM is in practice? I believe the proposed benefits can indeed be realized but it might be less than the theoretical maximum of 33% reported.

- Can you provide more details on how you perform the mid-training? How are the STEM embeddings initialized? How long does the loss spike (due to sudden architecture change) take to recover?

- Can you provide more downstream evaluations on more contextual reasoning heavy benchmarks? MMLU at the 1B scale seems barely above random guessing (25%) and moreover MMLU is also quite knowledge intensive.

---

> ### Author Response · Authors · 2025-11-23
> **Response to Reviewer YuJd (part 1/3)**
>
> We appreciate your supportive comments. We have provided practical wall-clock times for training, prefilling, and decoding for a fair evaluation of STEM's efficiency **(W1, Q1)**. In addition, we have provided additional benchmarking results to illustrate the superior contextual reasoning ability of STEM architecture compared to the dense baseline **(W2, Q3)**. We also clarify our midtraining setting and how the STEM model is initialized for this training stage **(Q2)**. We hope our detailed clarification with further experiment results will address your concerns.
> ### **(W1 & Q1) Can you provide some details on what the wall-clock time or GPU hour impact of STEM is in practice? I believe the proposed benefits can indeed be realized but it might be less than the theoretical maximum of 33% reported.**
> We thank the reviewer for raising this question. We want to note that our paper reports a **parameter saving of 33% in the FFN layers** and upto a **33% increase in training ROI**, defined as "Return on Investment" (Sec. 4, Page 5).
>
> **Throughput improvement:** Using our current implementation, we achieve **1.27x speedup in the MLP layers** which is close to the theoretical limit. In the table below, we provide the end-to-end speedups at 350M model scale using an NVIDIA H200 GPU (143.8 GiB HBM3e, 4.8 TB/s memory bandwidth, 600 W TDP).
>
> | Model | MLP Speedup | Training throughput (toks/s) | Prefilling Throughput (toks/s) |Decoding Throughput (toks/s) | Avg Downstream Accuracy (\%) |
> |----|------|------|----|-----|-----|
> |Baseline| - | 128K| 409K | 70.5K | 49.72 |
> |STEM    | 1.27x | 145K (1.13x) | 471K (1.15x) | 80.7K (1.14x) | 53.43 |
>
> > Note that at a 1.13x training speedup, we achieve average downstream task performance improvement of 3.6\%.
>
> Our current implementation gives a 1.13x improvement in training throughput, which translates to roughly 12\% savings in GPU hours. In the paper, we have elaborated on the theoretical speedup that can be expected (Sec 3.1, Page 4,5). As a follow-up, we shall work on improving our current implementation to match the expected speedups.
>
> **HBM Usage:** We only need to prefetch the embeddings with respect to the **unique tokens**. It is typically much smaller compared to the input activations multiplied with the gate projection outputs. Moreover, STEM eliminates the memory footprint of the up-projection parameters and their optimizer states. For additional speedup during decoding, we maintain an LFU cache of most frequently prefetched embeddings. This LFU cache size is strictly maintained to be smaller than the up-projection parameters.
>
> **Implementation details:** We keep the offloaded embedding table onto a *pinned* CPU memory. To improve the prefetching throughput, we implement **kernel-assisted copy**, that lets a CUDA kernel to fetch the required data from CPU with larger number of threads. This is especially beneficial when the retrieved data is scattered in the CPU memory. In addition, we use torch.compile to minimize CPU-side kernel launching overhead.

---

> ### Author Response · Authors · 2025-11-23
> **Response to Reviewer YuJd (part 2/3)**
>
> ### **(W2 & Q3) The concern about the hit to contextual reasoning abilities isn’t fully alleviated by the downstream numbers shown. MMLU and GSM8k are more reasoning heavy tasks but given the scale of the model, the improvement could be coming from the greater parametric knowledge capacity provided by STEM. Can you provide more downstream evaluations on more contextual reasoning heavy benchmarks? MMLU at the 1B scale seems barely above random guessing (25%) and moreover MMLU is also quite knowledge intensive.**
> We thank the reviewer for raising this concern. Interestingly, our additional experiments on more contextual reasoning-heavy tasks show that STEM's contextual reasoning skill is better than the dense baseline.
> To directly probe reasoning beyond parametric knowledge, we evaluate 1B-scale baseline and STEM models on **BIG-Bench Hard** (BBH)[1], **MuSR**[2], and the **LongBench**[3] *multi-hop reasoning* and *code-understanding* subsets. BBH is a collection of diverse, challenging tasks designed to require multi-step and compositional reasoning. MuSR requires the model under evaluation to track entities and constraints over a long narrative before answering a question. The LongBench multi-hop subset tests reasoning across multiple passages, while the code-understanding subset evaluates comprehension of complex code snippets. As shown in the table below, STEM consistently outperforms the dense baseline on BBH, MuSR, and on LongBench multi-hop and code-understanding tasks across all context-length ranges, indicating that STEM does not impair contextual reasoning and can in fact improve it.
> | Model    | BBH   |  MuSR |         LongBench            |           LongBench            |           LongBench            |           LongBench            | LongBench         | LongBench        |
> |----------|-------|-------------------------------|--------------------------------|--------------------------------|--------------------------------|-------------------|------------------|----|
> |          |       |   | Multi-hop <4k                 | Code <4k                       | Multi-hop 4–8k                 | Code 4–8k                      | Multi-hop ≥8k     | Code ≥8k         |
> | Baseline | 24.87 | 35.85 | 5.72                          | 45.37                          | 6.20                           | 44.64                          | 6.19              | 41.30            |
> | STEM     | 27.55 | 36.38 | 10.20                         | 52.68                          | 8.63                           | 52.53                          | 7.82              | 49.60            |
> We have included these additional benchmarking results in the appendix (Sec A.1, Page 15).
>
> [1] Challenging BIG-Bench Tasks and Whether Chain-of-Thought Can Solve Them
> (Suzgun et al, 2022)
>
> [2] MuSR: Testing the Limits of Chain-of-thought with Multistep Soft Reasoning (Sprague et al, 2024)
>
> [3] LongBench: A Bilingual, Multitask Benchmark for Long Context Understanding (Bai et al, 2024)
>
> ---
> ### **(Q2) Can you provide more details on how you perform the mid-training? How are the STEM embeddings initialized? How long does the loss spike (due to sudden architecture change) take to recover?**
> Our “mid-training” stage follows the standard continued-pretraining setup used in prior work [1,2]. Concretely, we first pretrain the STEM architecture and then resume training from this checkpoint on the mid-training corpus. Thus, the STEM embeddings are initialized directly from the existing pretrained checkpoint, and the architecture does not change at the mid-training boundary. As a result, we do not observe any loss spike associated with any sudden architectural change. We have updated our manuscript to clarify this sequential procedure (Sec 4.1, Page 7).
>
> We agree that another appealing setting is to start from a pretrained dense baseline and introduce STEM as a drop-in replacement just before midtraining. Exploring this regime is an interesting direction for future work. One possible strategy we are considering is to first collect up-projection outputs for different tokens and estimate principal directions for each token, then use these directions to initialize the corresponding STEM embeddings to ensure a smooth transition.
>
> [1] Does your data spark joy? performance gains from domain upsampling at the end of training (Blakeney et al, 2024)
>
> [2] 2 OLMo 2 Furious (OLMo Team, 2025)

---

> ### Author Response · Authors · 2025-11-23
> **Response to Reviewer YuJd (part 3/3)**
>
> ### **(S3) The paper performs an analysis of the pairwise cosine similarity of STEM embeddings vs original FFN up projection’s embeddings and finds that there is less redundancy in STEM embeddings. While this is an interesting finding, it is unclear whether reduced redundancy might negatively affect robustness.**
> We appreciate the concern raised by the reviewer. We consider that this reduced redundancy might not negatively affect robustness for the following reason. Under the key–value memory view [1] of transformer FFNs, STEM’s lower pairwise cosine similarity corresponds to a more incoherent dictionary of “value” vectors, and classical sparse-coding theory [2,3] shows that lower mutual coherence improves the stability of sparse representations under noise. Additionally, robustness studies frequently observe that adversarially robust models develop more diverse, less correlated features, and some defenses deliberately promote feature orthogonality/diversity as a robustness regularizer. Taken together, these results suggest that the reduced redundancy we observe in STEM embeddings is compatible with, and may even favor, robust representations. However, we intend to verify this hypothesis in our follow-up work.
>
> [1] Transformer Feed-Forward Layers Are Key-Value Memories (Geva et al, 2021)
>
> [2] Optimally sparse representation in general (nonorthogonal) dictionaries via $\ell1$ minimization (Donoho et al, 2003)
>
> [3] Greed is Good: Algorithmic Results for Sparse Approximation (Tropp et al, 2004)

---

### Author Response · Authors · 2025-11-23
**Response to all reviewers**

We thank reviewers [R1 (Yujd), R2 (8jLf), R3 (7Z8s), R4 (eAWT)] for their thoughtful and highly supportive feedback. We are glad that the reviewers found the problem statement **significant [R1, R3, R4]**, the observations and theoretical analysis **insightful and highly valuable [R1, R3, R4]**, the methods **simple, flexible, and effective [R1, R2, R3, R4]**, and our experiments and ablation studies **exhaustive [R2, R3, R4]**.

We have updated the paper to incorporate the constructive suggestions. We summarize the major changes below.

- **[R3, R4] Updated Table 3 caption, active parameter counts, and ROI metric**

  We have updated the caption of Table 3 (Page 7) to clearly describe the experimental setting. To clarify the distinction between **total parameters** and **active parameters** for different model variants, we add a column reporting active parameters per token. We also introduce a column for the ROI metric, which we now clearly define in the experiments section (Sec. 4, Page 5) as average accuracy divided by total training FLOPs.

- **[R1, R2, R4] Training and inference efficiency analysis**

  We present a more detailed analysis of training and inference efficiency (Sec. 3.1, Pages 4–5), including per-layer FLOPs, HBM usage and communication cost for both the dense baseline and STEM. We revised the exposition to keep the explanation as simple and readable as possible.

- **[R3] Better illustrative figures of architectural details**

  We have added additional figures describing the architectural design of STEM and its relationship to the standard gated FFN (Fig. 2, Page 3), to make the differences more visually clear.

- **[R3] Clearer explanation of technical details and embedding geometry**

  We have revised the analysis section (Sec. 3, Pages 3–4) to use more accessible language and to be easier to follow for a broader community of readers. Additionally, we clarify the rationale behind our STEM embedding geometry analysis (Sec. 5.1, Page 9), explicitly explaining why pairwise cosine similarity serves as a proxy for angular spread and mutual coherence.

- **[R1] Additional contextual reasoning–heavy benchmarks**

  We have added results on additional contextual reasoning–focused benchmarks, including BIG-Bench Hard, MuSR, and LongBench multi-hop and code-understanding subsets (Appendix Sec. A.1, Page 15). These results illustrate that STEM continues to outperform the dense baseline on challenging contextual reasoning tasks.

- **[R4] Additional long-context evaluation**

  We have added long-context evaluation results on the full LongBench test suite, stratified by context length, in addition to the synthetic NIAH task (Appendix Sec. A.2, Pages 15–16). These experiments support our hypothesis that STEM improves long-context performance relative to the dense baseline across a range of context lengths.

---

### Author Response · Authors · 2025-12-03
**A Call for AC's Discretion (Thanks to all the reviewers for your attention on STEM)**

We would like to express our whole-hearted gratitude to the reviewers who provided insightful comments and ACs for your additional efforts in moderating. We have tried to address the concerns raised by the reviewers, but it is unfortunate that we could not receive their valuable feedback because of unforeseen circumstances. However, to help our new AC peruse through our long paper and lengthy rebuttal history, we would like to summarize and organize the critical discussion points and the additional experiments we performed based on the comments received from our reviewers.


**1. Parameter matching and claimed performance gains [eAWT]**
We clarified in our response and our updated draft (Table 3, Page 8) that we keep the total number of **active parameters** the same across the standard transformer and the sparse alternatives. Additionally, to ensure a fair comparison between the sparse alternatives, our STEM architecture uses similar or fewer parameters in total compared to other sparse baseline like Hash MoE.

Moreover, we also highlight that, STEM (at 350M model scale) with half of the FFN layers replaced with STEM FFN, achieves a **4.48-point average accuracy improvement** over the dense baseline at **lower training FLOPs**, which is larger than the “3–4%” figure in the abstract. Across model sizes and training stages, STEM consistently improves the *average* downstream accuracy over the dense baseline.

**2. Additional contextual reasoning and long-context evaluations [Yujd, eAWT]**
To probe contextual reasoning, we added results on **BIG-Bench Hard**, **MuSR**, and **LongBench** (multi-hop and code-understanding subsets). STEM consistently outperforms the dense baseline on these reasoning-heavy tasks. Following R4[eAWT]’s suggestion, we also evaluated STEM on the full **LongBench** suite, stratified by context length. STEM shows **superior long-context performance** to the dense baseline across context length regimes, complementing our synthetic NIAH results.

**3. Training and inference efficiency analysis [8jLf]**
In Sec. 3.1 (Page 4), we added a more rigorous per-layer analysis of **training and inference efficiency**, deriving FLOPs and memory-access savings for STEM vs. the dense baseline across model sizes. The analysis also shows that STEM’s efficiency benefits are expected to become more pronounced at larger model scales.

**4. Concern about Wall-clock speedups [Yujd, 8jLf, eAWT]**
We complemented the theory with **practical throughput measurements**. For the 350M model, STEM achieves a **1.27× speedup in MLP layers** and end-to-end speedups of **1.13× (training)**, **1.15× (prefill)**, and **1.14× (decoding)** on H200 GPUs. We note that these numbers can likely be improved with further systems optimizations.
We want to note that our paper reports a parameter saving of 33% in the FFN layers and upto a 33% increase in training ROI, defined as "Return on Investment" (Sec. 4, Page 5). The expected end-to-end theoretical speedups are more clearly noted in our updated draft (Sec. 3.1, Page 4).


**5. Memory footprint, communication overhead, and scalability [8jLf]**
We quantified the additional memory needed for STEM embedding tables at larger model sizes and vocabularies, and showed it fits comfortably within the **CPU RAM** of standard public training nodes (e.g., AWS H200-based instances) while removing the up-projection weights and optimizer states from HBM. We also analyze CPU–GPU prefetching which is significantly optimized by **token deduplication**. In practice, token deduplication means we only prefetch embeddings for unique tokens, and the resulting STEM embedding prefetch accounts for roughly **10% of the per-step training latency**, which can be further hidden by overlapping computation and I/O. For decoding, we demonstrate that a GPU-resident **LFU cache** for frequent token embeddings keeps prefetch latency negligible relative to overall decoding time. Finally, we note that STEM’s communication/memory overhead scales roughly linearly with hidden size and depth, while overall training compute grows quadratically, indicating that STEM remains scalable at larger model sizes.

We are also grateful to **R3 [7Z8s]** for the constructive feedback about the presentation of our work. We have tried to address those concerns to the best of our effort.

We hope this summary helps clarify how the revised manuscript addresses the reviewers’ concerns on fairness of comparison, reasoning ability, long-context behavior, efficiency, and scalability.

---

### Meta-Review · Area_Chair_4ECu · 2025-12-27

**Summary:**

Reviewers found the simple, flexible. and insightful. In particular

- Reviewer YuJd found the idea interesting and raised questions about the FLOPs calculation and long-context understanding capabilities.
- Reviewer 8jLf thought the idea novel, and raised questions about the memory overhead because of the embedding lookup table and dimensionality ablation
- Reviewer 7Z8s raised concerns regarding the use of jargon/abbreviation and the lack of discussion compared to MOE
- Reviewer eAWT asked for a parameter-matched baseline and actual training/inference clock time.

Most of the concerns were addressed in rebuttal through quantitative results.

**Reviewer Concerns:**

The requested baseline, long-context evaluation, actual training/inference wall time and memory overhead were thorough discussed by the authors in the rebuttal.

**Reviewer Scores:**

Reviewers YuJd, 8jLf, 7Z8s would likely keep their original scores, and reviewer eAWT might increase the score.

---

### Decision · Program_Chairs · 2026-01-26

Accept (Poster)